# Spatial and stoichiometric in situ analysis of biomolecular oligomerization at single-protein resolution

Luciano A. Masullo [1,8] ✉, Rafal Kowalewski [1,2,8], Monique Honsa [1,2], Larissa Heinze[1,2], Shuhan Xu[1], Philipp R. Steen [1,2], Heinrich Grabmayr[1,2], Isabelle Pachmayr [1,3], Susanne C. M. Reinhardt [1,2], Ana Perovic[1], Jisoo Kwon[1], Ethan P. Oxley[4], Ross A. Dickins [4], Maartje M. C. Bastings [5], Ian A. Parish [6,7] & Ralf Jungmann [1,2] ✉

Latest advances in super-resolution microscopy allow the study of subcellular features at the level of single proteins, which could lead to discoveries in fundamental biological processes, specifically in cell signaling mediated by membrane receptors. Despite these advances, accurately extracting quantitative information on molecular arrangements of proteins at the 1–20 nm scale through rigorous image analysis remains a significant challenge. Here, we present SPINNA (Single-Protein Investigation via Nearest-Neighbor Analysis): an analysis framework that compares nearest-neighbor distances from experimental single-protein position data with those obtained from realistic simulations based on a user-defined model of protein oligomerization states. We demonstrate SPINNA in silico, in vitro, and in cells. In particular, we quantitatively assess the oligomerization of the epidermal growth factor receptor (EGFR) upon EGF treatment and investigate the dimerization of CD80 and PD-L1, key surface ligands involved in immune cell signaling. Importantly, we offer an open-source Python implementation and a GUI to facilitate SPINNA's widespread use in the scientific community.

Super-resolution microscopy has revolutionized our ability to investigate complex structures of intact cells with unprecedented resolution. State-of-the-art methods allow the study of subcellular features at the level of single proteins. This can be achieved by directly localizing molecules at sub-5 nm precision[1–3] or by applying computational algorithms to estimate proteins positions from sets of localizations at lower precisions (10–40 nm)[4,5].

DNA points accumulation for imaging in nanoscale topography (DNA-PAINT)[6,7], which uses the repetitive and transient binding of dye-labeled DNA oligos to their target-bound complements for super-resolution, is a robust method to translate localizations of fluorescent molecules to single proteins positions[8,9]. Furthermore, the recently developed Resolution Enhancement by Sequential Imaging (RESI)[10] extends DNA-PAINT to Ångström resolution, making it one of the preferred techniques to map the positions of single proteins in whole intact cells.

The direct visualization of the position of individual molecules has the potential to lead to significant advances in the understanding

[1]Max Planck Institute of Biochemistry, Planegg, Germany. [2]Faculty of Physics and Center for Nanoscience, Ludwig Maximilian University, Munich, Germany. [3]Department of Chemistry and Biochemistry, Ludwig Maximilian University, Munich, Germany. [4]Australian Centre for Blood Diseases, Monash University, Melbourne, VIC, Australia. [5]Institute of Materials and Interfaculty Bioengineering Institute, School of Engineering, École Polytechnique Fédérale de Lausanne, Lausanne, Switzerland. [6]Cancer Immunology Program, Peter MacCallum Cancer Centre, Melbourne, VIC, Australia. [7]Sir Peter MacCallum Department of Oncology, The University of Melbourne, Melbourne, VIC, Australia. [8]These authors contributed equally: Luciano A. Masullo, Rafal Kowalewski. ✉e-mail: masullo@biochem.mpg.de; jungmann@biochem.mpg.de

of fundamental biological processes, such as protein-protein interactions[9], signaling pathways[11], and receptor-ligand interactions[12], among others. Moreover, fundamental studies recently demonstrated how low-valency geometric patterns allow for super-selective targeting[13,14]. Downstream, the analysis of protein position maps could enable "spatial diagnostics" as a pre-screening method for personalized treatments and serve as a tool for biomedical discovery of patterned therapeutics[15,16] e.g. by guiding drug design principles[17].

However, despite the remarkable advances in super-resolution imaging, the quantitative analysis of acquired data remains a major challenge. Significant efforts have been undertaken to quantitatively analyze supramolecular, super-resolved structures in the 30–300 nm scale[18–21] analyzing the complete sets of localizations and treating the images as continuous structures. However, an analysis method to quantitatively evaluate the stoichiometry and spatial arrangement of discrete biomolecules in cells at the 1–20 nm scale (i.e., single-molecule level) is still missing.

Here, we introduce SPINNA (Single-Protein Investigation via Nearest-Neighbor Analysis): an analysis method that compares nearest-neighbor distances from experimental single-protein positions data with those obtained from simulated data based on a user-defined model of protein oligomerization states. It considers the heterogeneity of protein structures, random rotations, and labeling efficiencies[22] and uncertainties, allowing researchers to extract quantitative information about the stoichiometry, distribution, and arrangement of individual proteins into oligomers as well as their molecular compositions. We demonstrate the application of SPINNA in silico, in vitro, and in cells. As model systems, we investigate the molecular spatial arrangement of the epidermal growth factor receptor (EGFR) upon treatment with its ligand (EGF)[23,24], quantitatively assessing the resulting oligomerization. Furthermore, we investigate the dimerization of CD80 and the PD-1 ligand (PD-L1), two molecules regulating immune cell signaling[25,26]. Finally, we provide an open-source Python implementation and a Graphical User Interface (GUI) to maximize the immediate adoption and application of SPINNA within the scientific community.

## Results

### Concept and algorithm

SPINNA is a model-based analysis method that harvests the spatial information of the Nearest-Neighbour Distances (NNDs) of single biomolecules (e.g. proteins) detected in a cell. As such, it is a prerequisite to obtain a list of single-protein coordinates from repetitive localizations of fluorescent molecules, i.e. obtain a single-protein resolution image (Fig. 1a). The main conceptual steps of SPINNA can be summarized as follows:

**Model formulation.** The user creates a model that describes the multimer species possibly present in the cell by specifying the arrangement and the distances between the target protein(s) of

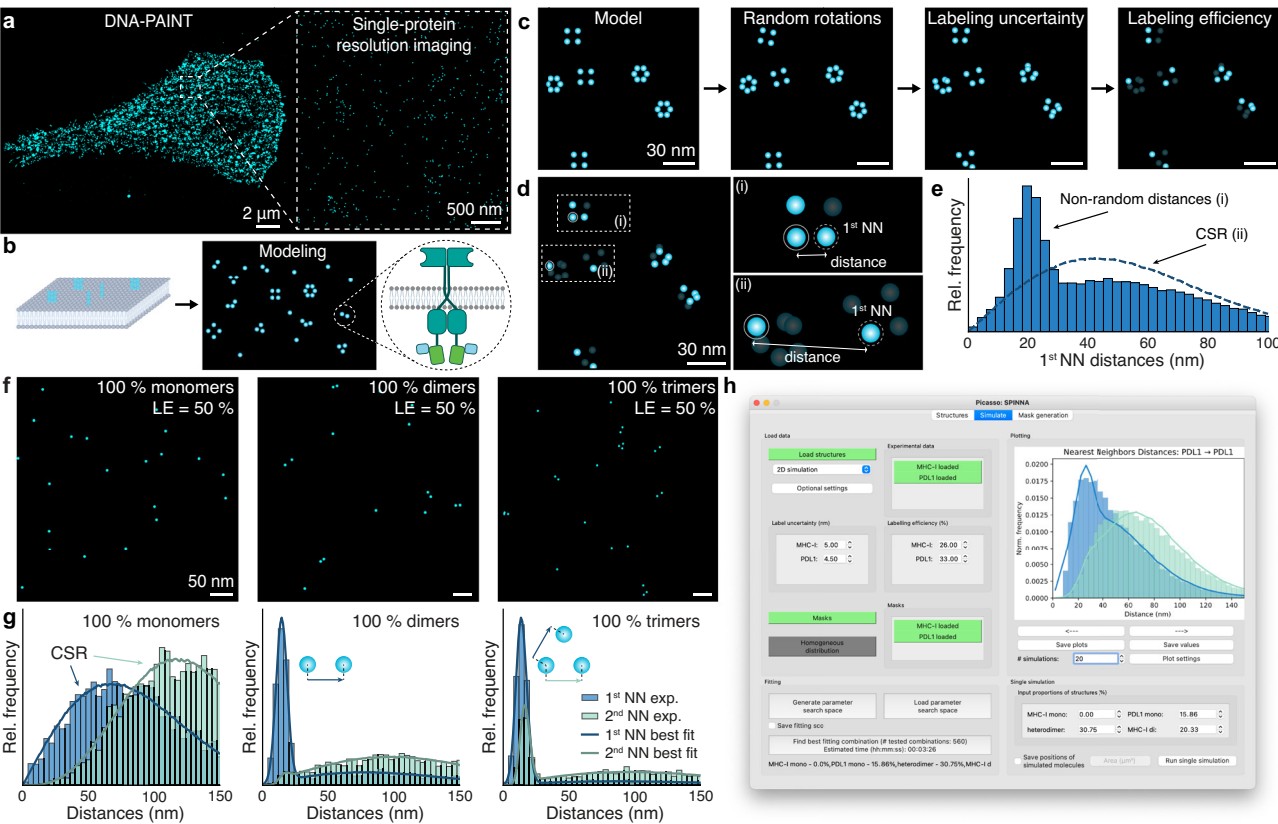

**Fig. 1 | SPINNA: concept, algorithm, and open-source software implementation. a** Current state-of-the-art super-resolution fluorescence microscopy such as DNA-PAINT or RESI enable single-protein resolution imaging in whole cells. **b** Individual proteins can form varied oligomeric structures that can be modeled with SPINNA. **c** SPINNA simulates observed single-protein positions in the following manner: the user defines the model of oligomeric structures simulated at random positions (first panel), followed by a random rotation of each structure (second panel). Then, uncertainty is applied to the positions of individual targets (third panel). Lastly, labeling efficiency is simulated (fourth panel). **d** Nearest-Neighbour Distances (NND) are calculated for each protein in the dataset. **e** Datasets including oligomers show distinct peaks that differ from the distribution corresponding to Complete Spatial Randomness (CSR). **f** In silico experiments illustrating the working principle of SPINNA for different simulated experimental situations. **g** NND histograms corresponding to the simulations displayed in **e**. Solid lines represent the best fits to the in silico data. Insets show how different structures impact the NND histograms. **h** Python open-source implementation and graphical user interface allow users to readily use SPINNA.

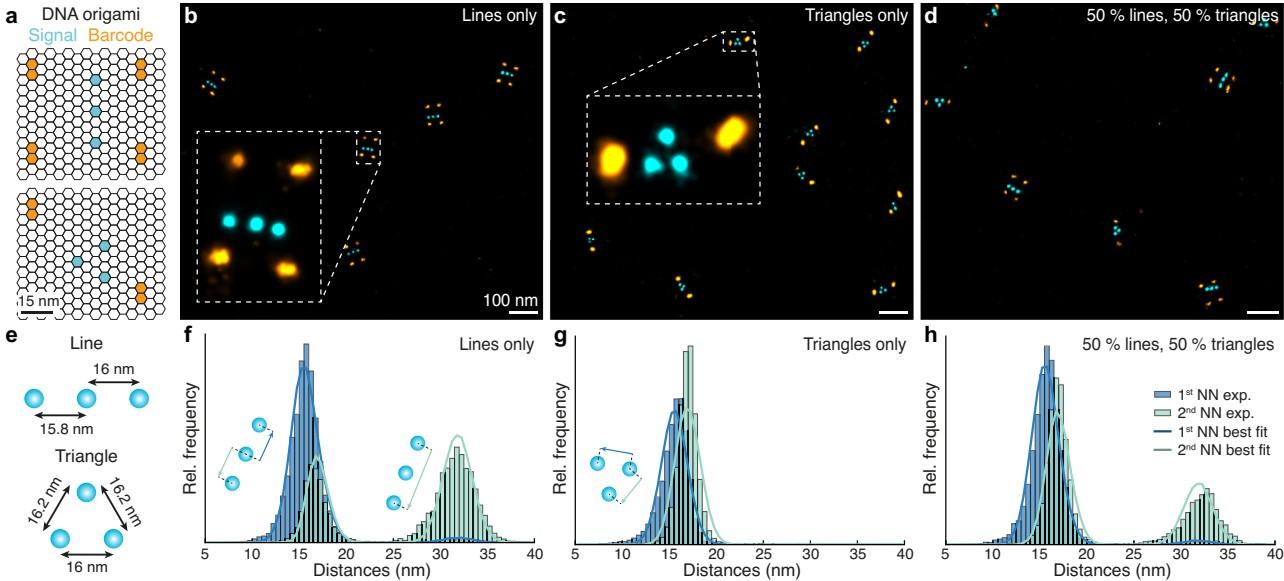

**Fig. 2 | Validation in vitro: DNA-origami nanostructures. a** Design of the two DNA-origami structures, emulating a linear and a triangular trimer respectively (light blue) and provided with a barcode for ground-truth identification (orange). **b** Exemplary field-of-view for the sample with linear nanostructures only. **c** Example of a field-of-view for the sample with triangular nanostructures only. **d** Exemplary field-of-view for the sample with a 1:1 mixture of linear and triangular nanostructures. **e** SPINNA model structures. **f** 1st and 2nd NND histograms for the sample with linear nanostructures only. SPINNA fits in blue and turquoise lines. Insets show how different structures impact the NND histograms. **g** Same as (**f**) for triangular nanostructures. **h** Same as (**f**) for the 1:1 mixture of linear and triangular nanostructures. Repeated three times with similar results.

interest (Fig. 1b). This step includes all the prior information about the system (e.g., from other measurements such as biochemical assays and protein structures). The position of the peaks in the NND histograms obtained from the experimental data can be used as an a priori estimate for determining the distances between proteins within the oligomer when there is limited structural information available on the target under investigation. The model should also include the potential bias and uncertainty associated with the labeling of the target (e.g. linkage error[27]). A simulated multimer is created by starting from the model, adding random rotations, labeling uncertainty, and accounting for labeling efficiency[22] as depicted in Fig. 1c.

**Labeling efficiency.** An accurate estimation of the labeling efficiency (ranging from 0 to 100%) of the target molecules must come from an independent experiment and can be obtained, for example, using recently developed quantitative approaches[22,28]. Labeling efficiency can have a significant impact on the accuracy of the results and should be carefully calibrated. For instance, if a labeling process (e.g. nanobody labeling) has an efficiency of 50% and the target molecules are arranged as tetramers, the most frequently observed structure will be an apparent dimer (Fig. 1c). The uncertainty in the determination of the labeling efficiency (e.g. 50 +/− 5%) will also have an impact in the results of the analysis that should be quantified (see Supplementary Data 1 and 2).

**Density distribution.** Heterogeneous protein density throughout the cell can be modeled by generating a density mask to account for larger-scale (>200 nm) density variation (see Supplementary Fig. 1). The density mask can also be omitted if no significant density variations are observed throughout the cell or if a sub-area with uniform density is selected for analysis.

**Search space.** Next, the user should define the range of parameters to be evaluated. The set of all the combinations of proportions of structures forms a search space of parameters over which the analysis will

be performed (see Supplementary Fig. 2). Additionally, label position uncertainty and intermolecular distances between structures can be added to the search space. The number of points in the stoichiometry search space (see "Granularity" in Methods) can be determined by the user.

**Simulation and NND calculation.** One simulated dataset is generated for a specified set of parameters and the NNDs are calculated (Fig. 1d, e). Datasets including oligomers show distinct peaks that differ from the distribution corresponding to Complete Spatial Randomness (CSR), i.e. uniform probability in space.

**Optimization procedure.** Finally, SPINNA creates a simulated dataset for each set of parameters in the search space and computes the best fit between the simulated and the experimental data by calculating the Kolmogorov-Smirnov 2-sample test statistic (KS2)[29] (see Methods). Figure 1f, g show examples of in silico experiments (simulated data, see Methods) and how different models fit the data. Additionally, bootstrapping can be used to find the uncertainty in the fitted stoichiometries (see Methods). KS2 values along with their uncertainties can be reported to compare the confidence between models.

We developed a Graphical User Interface (Fig. 1h) to ease the adaptation of SPINNA in a user-friendly fashion and make it accessible to non-experts in software programming. A detailed step-by-step user guide with practical examples is provided at the Zenodo repository https://doi.org/10.5281/zenodo.15131642.

### Experimental validation
**DNA origami.** As an experimental in vitro demonstration of SPINNA, we used self-assembled DNA origami structures to precisely position DNA binding sites[7,30]. We designed two DNA origami structures, containing three binding sites spaced approx. 15 nm, arranged in a line and a triangle respectively (Fig. 2a). The DNA origami structures are additionally equipped with a barcode set of binding sites that serve as a ground-truth identification, in this case, four and two pairs of binding

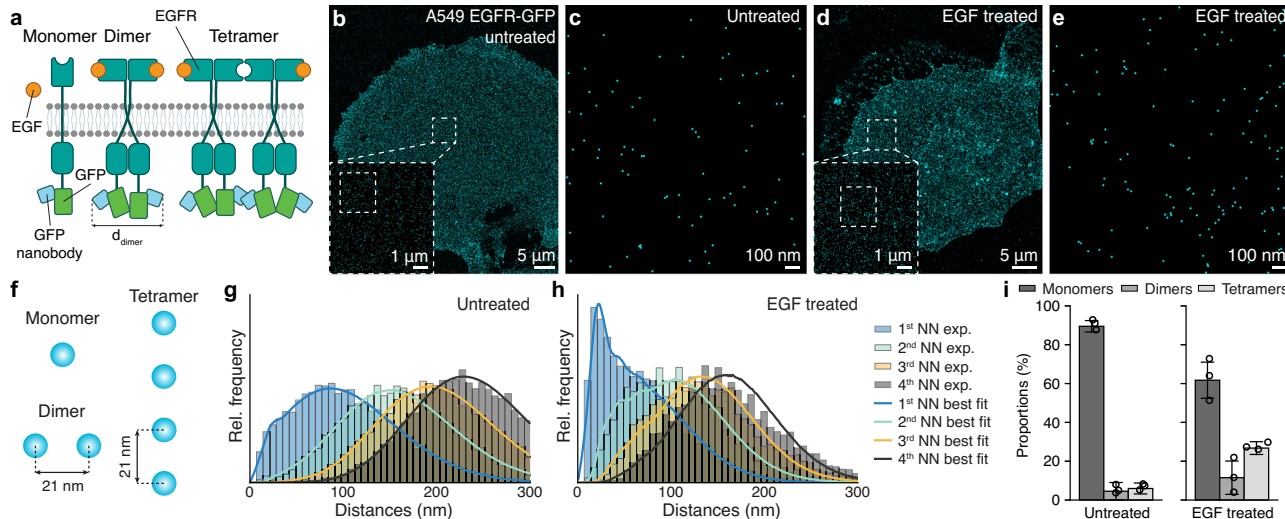

**Fig. 3 | Validation in cells: EGFR oligomerization upon EGF incubation.**
**a** Scheme of monomeric (left), dimeric (middle), and tetrameric (right) EGFR and the ligand, EGF. C-terminal-mEGFP and anti-GFP nanobody are also depicted.
**b** Overview of the whole A549 cell expressing EGFR-mEGFP without EGF treatment. **c** Zoom-in on a representative region for the untreated sample. **d** Same as (**b**) but for an EGF-treated cell. **e** Zoom-in on the sample treated with EGF for 10 min. **f** Model used in SPINNA consisting of a mixture of monomers, dimers, and

tetramers. **g** NND histograms and SPINNA fit for the untreated cell. **h** NND histograms and SPINNA fit for the EGF-treated cell. **i** Summarized results of SPINNA across 3 untreated ($n = 3$ technical replicates) and EGF-treated cells ($n = 3$ technical replicates). Stoichiometries are displayed by the bars (means), and error bars (standard deviation). A clear increase in oligomerized structures upon EGF treatment is observable.

sites for the line and the triangle origami, respectively. DNA origami structures represent a ground truth in vitro experiment to mimic oligomeric states of, for example, membrane receptors on the cell surface.

We then imaged three different samples containing: (i) only linear structures (Fig. 2b), (ii) only triangular structures (Fig. 2c), and (iii) a 1:1 mix of both linear and triangular structures (Fig. 2d). DNA origami structures were imaged in two rounds (one for the structures and one for barcodes) using Exchange-PAINT[31]. The localization precision achieved in these measurements was ~2 nm, allowing us to clearly resolve the individual binding sites, thus mimicking an experiment in cells with single-protein resolution. A SPINNA model consisting of triangular and linear trimers was created (Fig. 2e).

We obtained single binding site positions by clustering localizations as described before[10] and calculated 1st and 2nd NNDs for all binding sites. LE was estimated from the average number of detected binding sites (see Methods). We then performed SPINNA on each dataset and retrieved (i) $(95 \pm 1)$ % linear trimers for the lines-only sample (Fig. 2f), (ii) $(99 \pm 1)$ % triangular trimers for the sample containing only triangles (Fig. 2g), and (iii) $(54 \pm 1)$ % and $(46 \pm 1)$ % for the mix of lines and triangles, respectively (Fig. 2h), where the barcodes revealed a ground truth proportion of 50.3% and 49.7%. The following KS2 scores for the three datasets were obtained: $0.028 \pm 0.003$, $0.056 \pm 0.001$ and $0.039 \pm 0.002$, for the lines-only, triangles-only and mixed samples, respectively. We attribute the small deviations of the optimal fit from the ground truth to variations, flexibilities, and imperfections in the folding of some DNA origami that are not accounted for in our model. As a control, we show that adding other possible structures (e.g. monomers and dimers) to the model does not change the results (see Supplementary Fig. 3), demonstrating that SPINNA robustly and accurately estimates the ground-truth experimental situation. To further investigate the sensitivity and accuracy of SPINNA, we performed in silico experiments at different densities of molecules and numbers of detected molecules with structures equivalent to the ones of the DNA origami sample (Supplementary Fig. 4), showing that SPINNA yields accurate results within a large range of experimental conditions.

These experiments show not only that SPINNA can reveal molecular states in terms of stoichiometry (3 molecules per oligomer in this case) but also differentiate the geometrical arrangement of the oligomer (linear vs. triangular). This is a unique feature and distinguishes our approach from other fluorescence-based methods that can infer the number of (interacting) molecules but not their geometries[8,12]. Directly imaging and quantifying the geometric molecular arrangement of the target proteins is of utmost importance since it can play a critical role in the function and interactions of biomolecules[32].

**EGFR oligomerization.** To demonstrate the applicability of SPINNA in a cellular context, we next evaluated the oligomerization of the epidermal growth factor receptor (EGFR). This cell surface receptor plays a crucial role in the regulation of cell growth, survival, proliferation, and differentiation[33]. However, higher expression levels of EGFR or mutations in the EGFR gene are linked to the development and further progression of various cancers[24,34], and therefore, the EGFR signaling pathway has become a prominent cancer-treatment target[35].

EGFR is one of the four receptor tyrosine kinases in the ErbB family. It is reported that EGFR forms homodimers and heterodimers with other members of the ErbB family upon activation with growth factor ligands[36–38]. In the resting state, EGFR is present in the cell membrane mostly as a monomer. Upon ligand binding, the receptors can undergo a conformational change, leading to EGFR dimerization and ultimately forming higher-order oligomers[12] (Fig. 3a). One of the ligands inducing EGFR dimerization is the epidermal growth factor (EGF), which is a soluble ligand and has been shown to amplify the expression of EGFR[39,40]. Ligand-induced dimerization of the receptor starts the activation of the EGFR pathway[41] and it has already been shown that EGFR dimerization is strongly dependent on EGF concentration and incubation time[42,43]. However, until now, the oligomerization behavior of EGFR upon EGF activation has not been investigated with direct imaging at single-protein resolution.

Here, we visualized and quantified the oligomerization of EGFR upon EGF incubation treatment under physiological concentrations of EGF in cells[44]. A549 cells stably expressing mEGFP-EGFR were treated with 10 nM EGF for 10 min and were then fixed for imaging. Next,

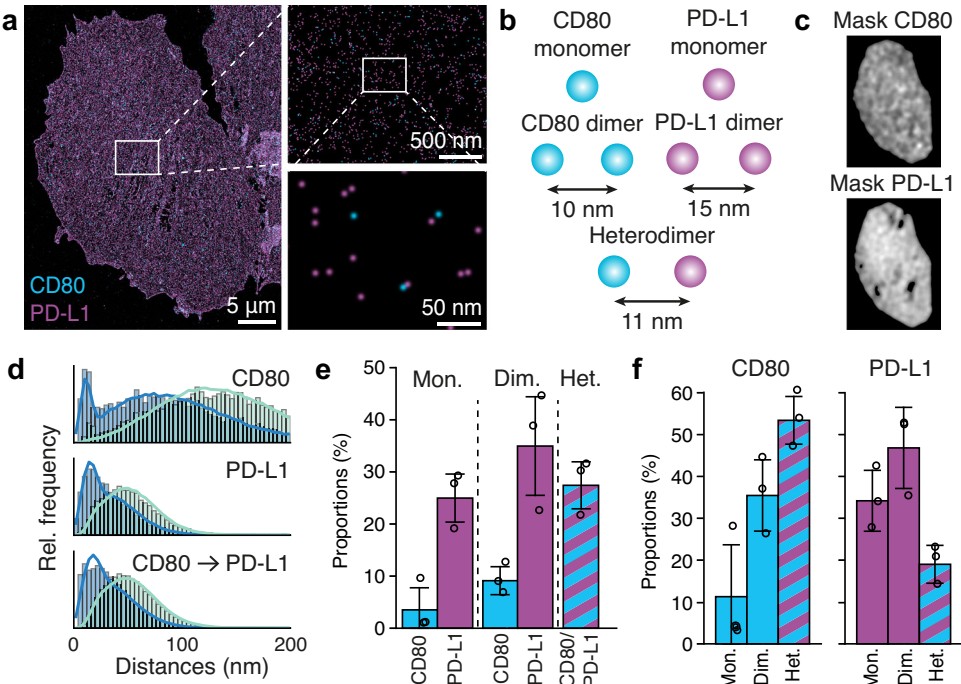

**Fig. 4 | CD80/PD-L1 dimerization. a** Overview of the entire CHO cell expressing CD80-mCherry (cyan) and PD-L1-mEGFP (magenta) and zoom-ins. **b** SPINNA model structures. **c** Density masks showing heterogeneous distributions of CD80 and PD-L1 in the cell from panel (**a**). **d** NND histograms and SPINNA fits. **e** Stoichiometries of monomers, homodimers, and heterodimers relative to the total number of molecules (independent of species). **f** Stoichiometries of monomers, homodimers, and heterodimers relative to the total molecules of each of the two species (CD80 and PD-L1) separately. Bars and error bars in (**e**) and (**f**) represent mean and standard deviation across $n = 3$ cells (technical replicates).

mEGFP-EGFR receptors were labeled with anti-GFP nanobodies (Fig. 3a), and whole cells were imaged using DNA-PAINT at ~3 nm localization precision (Fig. 3b–e) as described in the Methods. Areas in the cell membrane with homogeneous densities <50 µm⁻² were selected for analysis. At this resolution and density, single EGFRs could be clearly identified (Fig. 3c, e) and the oligomerization states assessed using SPINNA.

We created a model consisting of monomers, dimers, and tetramers with an intramolecular distance treated as a free parameter in the fit (Fig. 3f, and Methods). The best-fit distance was determined to be 21 nm. The model was fitted for both the untreated (Fig. 3g) and EGF-treated (Fig. 3h) cells, revealing stark differences in EGFR molecular organization. Figure 3i and Supplementary Data 1 display the obtained oligomer proportions ($N = 3$ cells per condition) indicating a significant oligomerization $(38 \pm 9)$ % upon EGF treatment as compared to $(11 \pm 4)$ % oligomerization for the untreated case. Furthermore, we validated the performance of SPINNA in silico (Supplementary Fig. 5) with realistic simulations to ensure that accurate results are retrieved for our range of experimental parameters.

**CD80/PD-L1 dimerization.** Next, we applied SPINNA to quantify immune cell membrane proteins CD80 and PD-1 ligand (PD-L1) dimerization at single-protein resolution. Targeted inhibition of the checkpoint molecule programmed cell death 1 (PD-1) can activate tumor-specific T cells[45] while enhancing PD-1 activity is expected to suppress autoreactive T cells and reduce autoimmune disease symptoms[26]. The binding of CD80 to PD-L1 in *cis* on primary activated dendritic cells prevents PD-L1 from interacting with PD-1 on T cells, which otherwise inhibits T cell activation[25]. Therefore, an in situ quantification at the molecular level of the dimerization of CD80 and PD-L1 is potentially highly relevant for developing therapeutic strategies targeting the PD-1 checkpoint.

Here, we visualized and quantified the dimerization of CD80 and PD-L1 in CHO cells stably transfected with previously described CD80-mCherry and PD-L1-mEGFP vectors[46]. Cells were seeded on glass slides and fixed for imaging. Next, CD80-mCherry and PD-L1-mEGFP were labeled with anti-mCherry and anti-GFP nanobodies respectively, whole cells were imaged using DNA-PAINT at ~3 nm localization precision (Fig. 4a), and single-protein positions were retrieved from the set of localizations (see Methods). A SPINNA model consisting of monomers, homo- and heterodimers was created (Fig. 4b). A distinct density mask was calculated for each protein species to account for larger-scale density variations throughout the cell surface (Fig. 4c). Then, the model was fitted to the experimental NNDs, and best-fitting parameters were obtained (Fig. 4d). Consistent proportions of PD-L1 and CD80 heterodimers $(27 \pm 5)$ % were observed in the cell membrane together with PD-L1 monomers $(25 \pm 5)$ %, PD-L1 homodimers $(35 \pm 11)$ %, and a smaller fraction of CD80 monomers $(4 \pm 3)$ % and homodimers $(9 \pm 3)$ % (Fig. 4e and Supplementary Data 2). On the other hand, by examining each molecular species individually (Fig. 4f and Supplementary Data 2), we observe that $(89 \pm 8)$ % of the CD80 and $(66 \pm 9)$ % of the PD-L1 molecules respectively are oligomerized in either homo- or heterodimers. It must be noted, however, that the densities of each molecular species in the transfected cell lines are different, yielding $142 \pm 39$ µm⁻² for CD80 and $409 \pm 46$ µm⁻² for PD-L1, which could affect the oligomerization ratios.

Our study demonstrates the capability of SPINNA to quantify oligomerization dynamics in a system where both CD80 and PD-L1 are co-expressed. While our results provide a proof-of-principle application of the method, we acknowledge that the expression levels of these molecules in our experimental setup may influence their oligomerization behavior. Future studies could explore the impact of more physiologically relevant expression levels by engineering cell lines with reduced PD-L1 expression. Additionally, experiments in which CD80 or

PD-L1 is expressed alone could further clarify their intrinsic oligomerization tendencies and the role of cis interactions.

## Discussion

We have developed an analysis framework for super-resolution images that enables spatial and stoichiometric analysis of biomolecular oligomerization in situ. Our technique, called SPINNA, can be readily applied to any super-resolution dataset in which single-protein resolution is achieved or positions of single proteins are estimated from the data, including DNA-PAINT measurements in which qPAINT[8] is leveraged[47]. To facilitate the rapid and easy implementation of our method, we provide an open-source Python software package and a graphical user interface.

SPINNA is a model-based method that allows not only the retrieval of the most likely oligomerization state (number of molecules) of the biomolecule(s) in the cell but also its spatial arrangement (geometry of the oligomer) which can be critical for the functionality of the molecular complex. For example, the interaction of receptor molecular complexes can change dramatically according to their spatial arrangement[17,48].

We first tested SPINNA in an in vitro experiment using DNA origami. SPINNA is able not only to recapitulate the stoichiometry of the measured oligomers but also to uniquely unveil their geometrical arrangement, in this case, a linear versus a triangular structure. This assay highlights the power of direct visualization and quantification of oligomerization states via fluorescence microscopy. We then visualized and quantified the oligomerization of the cancer therapeutic target EGFR in the resting state and upon binding to its ligand, EGF, in whole intact cells. We retrieved $(10 \pm 3)$ % of oligomerization for the resting state while we detected an oligomerization of $(38 \pm 9)$ % upon EGF treatment. Finally, we assessed the dimerization of PD-L1 and CD80, a relevant molecular interaction in dendritic cells' surface that plays a key role in T cell activation[25,26]. SPINNA reveals the heterodimerization of PD-L1 and CD80 $(27 \pm 4)$ % along with PD-L1 homodimerization $(34 \pm 7)$ % and homodimerization of CD80 $(11 \pm 3)$ %. When looking at each molecular species individually we observe oligomerizations of 93% (CI: 83–100%) and $(65 \pm 7)$ % for CD80 and PD-L1 respectively.

We have demonstrated both in vitro and in cellular experiments that SPINNA is a versatile tool that is able to quantify the (re)arrangement of biomolecular complexes in situ. We note that SPINNA can be readily extended to three-dimensional data and that different models including more complex molecular structures (e.g. flexible chains of oligomers with variable lengths, clusters of molecules) can be implemented. SPINNA is an analysis method that tests an experimental hypothesis summarized in the model. Therefore, it harvests prior information from previous experiments, such as biochemical assays[49], crystallographic[50] or cryoEM[51] structures, or AI-based predictions of molecular interactions[52].

For purely exploratory, hypothesis-free, scenarios, SPINNA should be complemented with other approaches, for example, those based on spatial clustering and point-pattern data mining. For example, properties of the NND distributions (e.g. peaks) can be used as an a priori estimate for interaction distances. Statistical functions like Ripley's K function could provide information on the sizes of the clusters and segmentation algorithms such as DBSCAN can then be used to define the molecular clusters, serving as an estimation for the degree of oligomerization. Also, geometrical descriptors (shape, concavity, linearity, circularity, etc.) of the clusters coming from the DBSCAN analysis could be used to point towards a suitable model to use in the downstream SPINNA analysis.

State-of-the-art super-resolution fluorescence microscopy methods such as DNA-PAINT[2,53] and RESI[10] allow resolving patterns of biomolecules at single-protein resolution with sufficient throughput (approx. 1–10 cells per hour of measurement) to address molecular cell biology questions thus far out of reach. SPINNA is poised to become the preferred analytical tool that will enable such molecular-resolution optical microscopy methods to be applied for 'spatial molecular diagnostics' as a pre-screening method for more accurate and personalized treatments and could also serve as a tool for biomedical discovery of patterned therapeutics—for example, by guiding drug design principles and analyzing drugs modes of action at the molecular scale.

## Methods

### Materials

Unmodified DNA oligonucleotides were purchased from MWG Eurofins and Metabion. DNA oligonucleotides modified with C3-azide and Cy3B were ordered from Metabion and MWG Eurofins. M13mp18 and p7560 scaffolds were purchased from Tilibit. Magnesium chloride (1 M; AM9530G), sodium chloride (5 M; AM9759), ultrapure water (10977-035), Tris (1 M, pH 8; AM9855G), EDTA (0.5 M, pH 8.0; AM9260G) and 10× PBS (70011051) were purchased from Thermo Fisher Scientific. BSA (A4503-10G) was ordered from Sigma-Aldrich. Triton X-100 (6683.1) was purchased from Carl Roth. Sodium hydroxide (31627.290) was purchased from VWR. Paraformaldehyde (15710) and glutaraldehyde (16220) were obtained from Electron Microscopy Sciences. Tween-20 (P9416-50ML), glycerol (65516-500 ml), methanol (32213-2.5 L), protocatechuate 3,4-dioxygenase pseudomonas (PCD; P8279), 3,4-dihydroxybenzoic acid (PCA; 37580-25G-F) and (±)-6-hydroxy-2,5,7,8-tetra-methylchromane-2-carboxylic acid (Trolox; 238813-5 G) were ordered from Sigma-Aldrich. Neutravidin (31000) was purchased from Thermo Fisher Scientific. Biotin-labeled BSA (A8549) and Sodium azide (769320) were obtained from Sigma-Aldrich. Coverslips (0107032) and glass slides (10756991) were purchased from Marienfeld and Thermo Fisher Scientific. Double-sided tape (665D) was ordered from Scotch. FBS (10500-064), 1× PBS (pH 7.2; 20012-019), 0.05% trypsin–EDTA (25300- 054), Salmon Sperm DNA (15632011), OptiMEM (31985062) and Lipofectamine LTX (A12621) were purchased from Thermo Fisher Scientific. Ninety-nanometer gold nanoparticles (G-90-100) were ordered from Cytodiagnostics. Nanobodies against GFP (clone 1H1) with a single ectopic cysteine at the C-terminus for site-specific conjugation were purchased from Nanotag Biotechnologies. DBCO-PEG4-Maleimide (CLK-A108P) and DBCO-AF647 (CLK-1302A) were purchased from Jena Bioscience.

### Buffers

The following buffers were used for sample preparation and imaging:
- DNA origami folding buffer: 10 mM Tris, 1 mM EDTA, 12.5 mM MgCl$_2$, pH 8
- FoB5 buffer: 5 mM Tris, 1 mM EDTA, 5 mM NaCl, 5 mM MgCl$_2$, pH 8
- Buffer A: 10 mM Tris pH 8, 100 mM NaCl, and 0.05% Tween-20
- Buffer B: 10 mM MgCl$_2$, 5 mM Tris-HCl pH 8, 1 mM EDTA, and 0.05% Tween-20, pH 8
- Buffer C: 1× PBS, 1 mM EDTA, and 500 mM NaCl, pH 7.4; 0.02% Tween; optionally supplemented with 1× Trolox, 1× PCA and 1× PCD
- Blocking buffer: 1× PBS, 1 mM EDTA, 0.02% Tween-20, 0.05% NaN$_3$, 2% BSA, 0.05 mg/ml sheared salmon sperm DNA
- Quenching buffer: 2 M NH$_4$Cl in ddH$_2$O

**PCA, PCD, and Trolo.** 100× Trolox was made by adding 100 mg of Trolox to 430 µl of 100% methanol and 345 µl of 1 M NaOH in 3.2 ml water. 40× PCA was made by mixing 154 mg PCA in 10 ml water and NaOH and adjusting the pH to 9.0. 100× PCD was made by adding 9.3 mg PCD to 13.3 ml of buffer (100 mM Tris-HCl pH 8, 50 mM KCl, 1 mM EDTA, 50% glycerol).

**DNA origami self-assembly.** All DNA origami structures were designed in Picasso Design[7]. Self-assembly of DNA origami was

accomplished in a one-pot reaction mix with a total volume of 40 µl, consisting of 10 nM scaffold strands (for sequence, see Supplementary Data 3), 100 nM folding staples (Supplementary Data 4), 500 nM bio-tinylated staple strands (Supplementary Data 4) and 1 µM staple strands with docking site extensions (Supplementary Data 4) in DNA origami folding buffer. The reaction mix was then subjected to a thermal annealing ramp using a thermocycler. First, it was incubated at 80 °C for 5 min, cooled using a temperature gradient from 60 to 4 °C in steps of 1 °C for 3.21 min, and finally held at 4 °C. DNA origami featuring three R1-docking sites arranged in a triangle alongside two R3-barcode positions and DNA origami featuring three R1-docking sites arranged in a line alongside four R3-barcode positions were folded.

**DNA origami purification.** DNA origami structures were purified via ultrafiltration using Amicon Ultra Centrifugal Filters with a 100-kDa molecular weight cutoff (MWCO; Merck Millipore, UFC510096) as previously described[54]. Folded origamis were brought to 500 µl with FoB5 buffer and spun for 3.5 min at 10,000 × g. This process was repeated twice. Purified DNA origami structures were recovered into a new tube by centrifugation for 5 min at 5000 × g. Purified DNA origami structures were stored at −20 °C in DNA LoBind tubes (Eppendorf, 0030108035).

**DNA-PAINT docking and imager strand sequences.** Two orthogonal DNA sequence motifs were used as the binding sites for the main (Fig. 2a, light blue) and barcode (Fig. 2a, orange) imaging rounds, respectively. The docking strands were 5xR1 (TCCTCCTCCTCCTCCTCCT) and 7xR3 (CTCTCTCTCTCTCTCTCTCT). The respective imagers were R1 (AGGAGGA-Cy3B) and R3 (GAGA-GAG-Cy3B).

**Microscope setup.** Fluorescence imaging was carried out on an inverted microscope (Nikon Instruments, Eclipse Ti2) with the Perfect Focus System, applying an objective-type TIRF configuration equipped with an oil-immersion objective (Nikon Instruments, Apo SR TIRF×100, NA 1.49, Oil). A 560-nm laser (MPB Communications, 1 W) was used for excitation and coupled into the microscope via a Nikon manual TIRF module. The laser beam was passed through a cleanup filter (Chroma Technology, ZET561/10) and coupled into the microscope objective using a beam splitter (Chroma Technology, ZT561rdc). Fluorescence was spectrally filtered with an emission filter (Chroma Technology, ET600/50 m and ET575lp) and imaged on an sCMOS camera (Hamamatsu Fusion BT) without further magnification, resulting in an effective pixel size of 130 nm after 2 × 2 binning. TIR illumination was used for all measurements. The central 1152 × 1152 pixels (576 × 576 after binning) of the camera were used as the region of interest. The scan mode of the camera was set to "ultra quiet scan" (readout noise = 0.7 e- r.m.s., 80 µs readout time per line). Raw microscopy data was acquired using µManager (Version 2.0.1)[55].

**DNA origami sample preparation and imaging.** For sample preparation, a bottomless six-channel slide (Ibidi, no. 80608) was attached to a coverslip. First, 200 µl of biotin-labeled BSA (1 mg/ml, dissolved in buffer A) was flushed into the chamber and incubated for 2 min. The chamber was then washed with 200 µl of buffer A. Then, a volume of 200 µl of neutravidin (0.1 mg/ml, dissolved in buffer A) was flushed into the chamber and incubated for 5 min. After washing with 200 µl of buffer A and subsequently with 200 µl of buffer B, 60 µl of biotin-labeled DNA structures (concentration 200 pM) in buffer B was flushed into the chamber and incubated for 6 min. After DNA origami incubation the chamber was washed 3× with 200 µl of buffer B. Finally, 200 µl of the imager solution in buffer B was flushed into the chamber. The chamber remained filled with imager solution and imaging was then performed. Between imaging rounds, the sample was washed three times with 1 ml of buffer B until no residual signal from the

previous imager solution was detected. Then, the next imager solution was introduced. Imaging was performed in TIRF, using 45 mW of 560 nm laser power measured after the objective, corresponding to a power density of 225 W/cm². Imager concentration was 1 nM for both R1 and R3 imagers and the measurement length for every round was 1 hour (36,000 frames at 100 ms per frame).

**CHO cell line engineering.** CHO cells were co-transduced with two retroviral vectors: one encoding CD80 fused to intracellular mono-meric Cherry (mCherry) and another encoding PD-L1 fused to an intracellular monomeric green fluorescent protein (mGFP). A single-cell clone was isolated and expanded as described in previous work[46].

**Cell culture.** A549 EGFR-GFP cells (Sigma-Aldrich: CLL 1141) and CHO cells were cultured at 37 °C and 5% CO₂ in DMEM medium (Gibco, no.61965026) supplemented with 10% FBS (Gibco, no. 11573397). Cells were passaged every 2–3 days using trypsin-EDTA (Gibco, no. 25300096).

**Nanobody-DNA conjugation.** The anti-GFP nanobody (clone 1H1, Nanotag Biotechnologies) and the anti-mCherry nanobody (clone 2B12, Nanotag Biotechnologies) were conjugated to a DBCO-PEG4-Maleimide linker (no. CLK-A108P, Jena Bioscience). After removing unreacted linker with Amicon centrifugal filters (10,000 MWCO), the DBCO-nanobody was then conjugated via DBCO-azide click chemistry to either the docking strand 5'- azide - TCC TCC TCC TCC TCC TCC T -3' (Metabion) or the docking strand 5'- azide -ACC ACC ACC ACC ACC ACC A -3'. A detailed description of the DNA conjugation to the nanobody can be found in[2].

**Expression and purification of EGF.** GST-GSGS-GB1-TEV-GGGSGGG-hEGF was expressed in Escherichia coli T7 SHuffle Express strain. After the expression, the cells were harvested and lysed by sonication. The lysate was clarified by centrifugation at 50,000 × g for 30 min at 4 °C. For GST Affinity Purification, the clarified lysate was loaded onto a Glutathione Sepharose 4B column equilibrated with 50 mM Tris-HCl (pH 7.5) and 200 mM NaCl. After washing with 10 column volumes (CV) of binding buffer, GGGSGGG-hEGF was eluted by TEV protease clea-vage (1 mg/ml in 50 mM Tris-HCl, 200 mM NaCl) overnight at 4 °C. The cleaved protein was further purified using a Superdex 30 Increase GL 10/300 column equilibrated with 50 mM Tris-HCl (pH 7.5) and 150 mM NaCl. Fractions containing the purified GGGSGGG-hEGF were pooled. The purified protein samples were analyzed via SDS-PAGE, using 4–22% gradient gels under non-reducing conditions.

**Sortase-mediated conjugation of GGGSGGG-EGF.** Site-specific functionalization of GGGSGGG-hEGF was achieved by reacting 100 µM of the protein with 500 µM of an Azide-peptide (Azide-LPETGG-HHHHHH), in the presence of 10 µM Sortase 7 M in 50 mM Tris-HCl (pH 7.5), 150 mM NaCl. The reaction was carried out at 16 °C for 4 h. The resulting conjugate was purified by passing the reaction mixture through 200 µl of Ni-INDIGO Agarose to remove the His-tagged Sortase and any unreacted peptide. The Azide-EGF was further conjugated with DBCO-AF647. A reaction mixture containing 40 µM of Azide-EGF was incubated overnight at 4 °C with an equimolar amount of DBCO-AF647. The conjugated product AF647-EGF was con-centrated to a final concentration of 8.5 µM using Amicon Ultra 3 kDA filters.

**EGFR-mEGFP imaging.** 10,000 cm⁻² A549 GFP-EGFR cells were seeded on eight-well high glass-bottom chambers (Ibidi, no. 80807). The next day, the cells were washed three times with serum-free DMEM medium and starved for 6 h. Next, the cells of the "treated" experiment were treated for 10 minutes with 10 nM EGF in a serum-free DMEM medium. The "untreated" cells were not treated with EGF. The cells in both

experiments were then fixed with pre-warmed methanol-free 4% PFA (Thermofisher, no. 043368.9 M) in 1× PBS for 15 min. After washing 3 times with 1× PBS, the cells were permeabilized with 0.125% TritonX-100 (Sigma Aldrich, no. 93443) in 1× PBS for 2 min. After washing three times with 1× PBS, the cells were blocked with blocking buffer at 4 °C overnight. 25 nM DNA-conjugated anti-GFP nanobodies in blocking buffer were incubated for 1 h at RT. The cells were washed three times with 1× PBS, post-fixed with 4% PFA and 0.2% glutaraldehyde (Serva, no. 23115.01) in 1× PBS for 10 min. Then, the sample was quenched with freshly prepared 200 mM quenching buffer in 1× PBS for 5 min and washed three times with 1× PBS. 90 nm gold nanoparticles (Absource, no. G-90-100) diluted 1:1 in 1× PBS were incubated for 5 min at RT. The cells were washed three times with 1× PBS. The imaging was performed with 200 pM imager strand (5′-AGG AGG A-Cy3B-3′ or 5′-TGG TGG-Cy3B-3′, obtained from Metabion) in Buffer C. In each field of view, 40,000 frames with 100 ms exposure time per frame were acquired. A laser power of 30 mW (560 nm, measured after the objective) was used, corresponding to a power density of ~150 W/cm².

**mEGFP-PD-L1 and mCherry-CD80 imaging.** 30,000 cm$^{-2}$ CHO cells were seeded in a well on a coverslip (Ibidi, Cat.No: 80827) and cultured for 24 h at 37 °C. Subsequent procedures are all carried out at room temperature and pressure. The cells were fixed for 15 min with 4% PFA prewarmed to 37 °C, permeabilized with 0.1% Triton X-100 for 5 min, and blocked for 60 min with blocking buffer. Three times washing with 1 × PBS was carried out between each step. Then they were stained with anti-GFP and anti-mCherry nanobodies (clone 1H1 and 2B12 from NanoTag Biotechnologies) at 50 nM each in blocking buffer for 60 min. Anti-GFP nanobody is conjugated with the R4 sequence and anti-mCherry nanobody is conjugated with the R3 sequence. The cells were washed three times with washing buffer and incubated in Buffer C for 5 min. Afterward, they were postfixed with 2% PFA + 0.2% glutaraldehyde for 30 min and quenched with 100 mM NH$_4$Cl for 20 min. Before imaging, the coverslip was incubated with 90 nm gold nanoparticles for 5 min. The cells were washed 3 times with 1x PBS. Imaging was conducted in TIR illumination using a two-round Exchange-PAINT protocol. The imaging buffer was Buffer C. For CD80-mCherry imaging, 200 pM R3 imager (sequence: AGAGAGA-Cy3b) solution was used and 75,000 frames of 100 ms integration time was collected per field of view; for PD-L1-mEGFP imaging, 100 pM R4 imager (sequence: TGTGTGT-Cy3b) was used and 150,000 frames of 100 ms integration time was collected per field of view. A laser power of 30 mW (560 nm, measured after the objective) was used, corresponding to a power density of ~150 W/cm².

**Single-molecule localization analysis.** Raw fluorescence data were reconstructed using the Picasso software package[7] (the latest version is available at https://github.com/jungmannlab/picasso). Drift correction was performed with a redundant cross-correlation and gold particles as fiducials for cellular experiments. Next, in the case of two-channel data, channels were aligned to each other using redundant cross-correlation. Subsequently, localizations were processed using the Picasso clustering algorithm. Circular clusters of localizations centered around local maxima were identified and grouped (assigned a unique identification number). Subsequently, the centers of the localization groups were calculated as weighted mean by employing the squared inverse localization precisions as weights.

**DNA origami barcodes.** Triangle DNA origami structures were equipped with 2 extra sets of binding sites, distanced at 72 nm. Line DNA origami structures were equipped with four extra sets of binding sites, arranged in a rectangular shape, where the distances between the sets of binding sites were 40 nm, 60 nm, and 72 nm (see Fig. 2a). To extract the identity of the DNA origami, the observed structures were

picked using Picasso: Render's "Pick similar" and the barcode localizations were clustered as described above with a radius of 6 nm and 15 minimum localizations. If two or four sites were detected and the distances between the sites matched the ground-truth distances within a 5 nm margin of error, the structures were assigned their corresponding identity.

**Parameter fitting with SPINNA.** To assess the fit between experimental and simulated data, we used the two-sample Kolmogorov-Smirnov (KS2) test on nearest neighbor distances (NNDs). For each comparison, the 1st NNDs of both datasets were used as input to calculate the KS2 test statistic. The test was then applied sequentially to the 2nd, 3rd, and subsequent higher-order NNDs, stopping at the highest NND automatically defined by the SPINNA model according to the maximum oligomer order. If the oligomer was $n$-meric, NNDs up to the (n-1)th order were used, as the nth NND is expected to follow a CSR distribution. We then reported the average KS2 test value across these NNDs. Users may also manually specify the highest NND order to consider. The parameters of the model that produced the best KS2 metric were selected as the optimal fit parameters.

**Granularity.** It determines the "percentage steps" (e.g. steps of 1 %pt) for the proportions of oligomers that will be used to create the search space and thus it will limit the resolution in terms of oligomers proportions that can be evaluated, i.e. the computational resolution. See Supplementary Fig. 2 for details. When the search space is created and saved, the user can examine what the different resulting percentage steps are for every tested oligomeric structure and thus choose the appropriate granularity. In principle, higher granularity does not have any disadvantage and always a sufficiently high granularity should be used. The only disadvantage is the computational cost that scales linearly with the amounts of points in the search space (numerical simulations performed). The number of points in the search space and the estimated total calculation time are displayed before the user decides to start the analysis. We recommend using a granularity that results in a "percentage step" that is substantially smaller than the smallest uncertainty coming from experimental sources (most importantly cell-to-cell variability). In concrete terms, we believe that a granularity resulting in percentage steps of 1 to 3 %pt should be a robust choice for most common scenarios, but this should be evaluated on a per-case basis.

**SPINNA in silico experiments.** For the examples displayed in Fig. 1f, g, three datasets were simulated with label position uncertainty of 3 nm and labeling efficiency of 50% with only monomers, dimers, and trimers, respectively. SPINNA was run with the three model structures and the same simulation parameters as above. Granularity was set to 51 (proportions iterated by 2 %pt). The following proportions were recovered (dataset/recovered proportions): (monomers only/100% monomers), (dimers only/98% dimers, 2% trimers), (trimers only/98% trimers, 2% dimers).

**SPINNA on DNA origami.** To account for deviations from the design parameters (15 nm distances), the value of the distances between sites of the origami were left as free parameters of the model and swept for best fit. For the line structure, inter-site distances were swept in the range of 15 nm to 16.5 nm with a step size of 0.1 nm. For the triangle structure, an isosceles triangle was assumed, with its base and sides ranging from 15 to 16.5 nm with a step size of 0.1 nm. Label position uncertainties were tested in the range from 0.8 nm to 1.4 nm with a step size of 0.1 nm. Using the lines- or triangles-only datasets as ground truth, the corresponding stoichiometries were simulated and the resulting KS2 values were reported. The combination of the four structure parameters yielding the lowest KS2

value, as depicted in Fig. 2e, and label position uncertainty of 1.1 nm were found to fit the experimental data the best. To find labeling efficiency for each dataset, the average fraction of detected sites was calculated, which ranged between 92.8% and 95.3%. 250,000 sites were simulated in SPINNA. Granularity was set to 101, such that the proportions of line and triangle structures were iterated by 1 %pt.

**SPINNA on EGFR.** Areas in the cell membrane with homogenous density were selected for the analysis. Dimer and tetramer intermolecular distance and label position uncertainty were assumed constant across oligomers, left as a free parameter of the model and determined from the fit. The distance ranged from 16 nm to 22 nm with a step size of 1 nm and label position uncertainties ranged from 4.5 nm to 7.0 nm with a step size of 0.5 nm. These ranges of values were chosen according to previous values reported in literature[56]. KS2 was reported for each set of parameters and the lowest KS2 value was found for a distance of 21 nm and label position uncertainty of 6.0 nm. Granularity was set to 34, i.e., proportions of structures were iterated by 3 %pt. Labeling efficiency of 37% was used, as reported in ref. 22.

**SPINNA on CD80/PD-L1.** To determine the homodimer intermolecular distance as well as the label position uncertainty for CD80, SPINNA was run on the CD80 only dataset with a distance ranging from 7 nm to 13 nm with a step size of 1 nm and the label position uncertainty ranging from 2 nm to 7 nm with a step size of 0.5 nm. The analogous procedure was conducted for PD-L1, with the distance ranging from 10 nm to 18 nm with a step size of 1 nm and the label position uncertainty ranging from 4 nm to 9 nm with a step size of 0.5 nm. Once the four parameters were established, the analogous pipeline was run to find the heterodimer intermolecular distance with test values ranging from 10 nm to 20 nm with a step size of 1 nm. These ranges of values were chosen according to previous values reported in literature[57–59]. Labeling efficiency values were used according to previous work[22]. Granularity was set to 16. Masks were generated by applying Gaussian blur with σ of 500 nm to molecule positions histogrammed with a bin size of 50 nm.

**SPINNA fit uncertainty estimation.** Bootstrapping was used to determine the uncertainties in the stoichiometries fitted by resampling the model based on the best-fitted stoichiometries[60]: For N experimentally observed molecules, N molecules were simulated in the SPINNA-found proportions of structures. SPINNA was run on simulated data and the fitted stoichiometries were saved. This procedure was repeated 10–20 times, depending on N. The mean values of fitted proportions of structures were reported, with their standard deviation as uncertainty. Additionally, KS2 values can be reported for each of the bootstrap fits, such that the user can compare the variations in improvement when testing different models.

### Reporting summary
Further information on research design is available in the Nature Portfolio Reporting Summary linked to this article.

## Data availability
The single-molecule localization data generated in this study have been deposited in the Zenodo database, https://doi.org/10.5281/zenodo.15131642

## Code availability
Raw image processing and SPINNA can be performed using Picasso available via GitHub at https://github.com/jungmannlab/picasso. SPINNA's documentation is available at https://picassosr.readthedocs.io/en/latest/spinna.html.

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

## Acknowledgements

This research was funded in part by the European Research Council through an ERC Consolidator Grant (ReceptorPAINT, grant agreement number 101003275), the BMBF (Project IMAGINE, FKZ: 13N15990), the Volkswagen Foundation through the initiative 'Life?—A Fresh Scientific Approach to the Basic Principles of Life' (grant no. 98198), the Danish National Research Foundation (Centre for Cellular Signal Patterns, DNRF135), the Max Planck Foundation and the Max Planck Society. We thank the MPI of Biochemistry core facility for the production of Sortase 7M as well as for azide-peptide synthesis. L.A.M. acknowledges a postdoctoral fellowship from the European Union's Horizon 20212022 research and innovation program under Marie Skłodowska-Curie grant agreement no. 101065980. R.K., M.H., S.X, P.R.S., I.P., S.C.M.R. acknowledge support by the IMPRS-ML graduate school. E.P.O and R.A.D. are supported by a CUREator grant from the Australian Government Medical Research Future Fund. I.A.P. is supported by the Victorian Cancer Agency Mid-Career Fellowship 21019. Parts of the illustration in Fig. 1b were created using Biorender.

## Author contributions

L.A.M. and R.K. conceived the analysis method, developed the algorithms, and implemented the software. L.A.M., M.H., L.H., S.X., and P.R.S. performed the experiments. L.A.M. and R.K. analyzed the data. H.G., I.P., and S.C.M.R. contributed to the analysis methods. I.P., J.K., and A.P. contributed to sample preparation. E.P.O. and R.A.D. developed the CHO stably transfected cell line. L.A.M., R.K., and R.J. interpreted the data and wrote the manuscript with input from all authors. L.A.M. and R.J. conceived and supervised the project. M.M.C.B. and I.A.P. contributed to the concept idea. L.A.M. and R.K. contributed equally.

## Funding

## Competing interests

The authors declare no competing interests.
