## [Transparent Peer Review file · Nature Communications]

Spatial and stoichiometric in situ analysis of biomolecular oligomerization at single-protein resolution

Corresponding Author: Professor Ralf Jungmann

Version 0:

Reviewer comments:

Reviewer #1

(Remarks to the Author)

In this manuscript, Masullo et al. report on SPINNA (Single-Protein Investigation via Nearest-Neighbor Analysis), a model-based method capable of determining the stoichiometry and the spatial arrangement of molecular targets within cells from single-protein resolution images, as well as calculating the percentage of molecular targets that adopt different geometrical configurations (i.e., linear vs triangular, dimers vs tetramer, etc). The authors benchmark the method's capabilities using both simulated data sets and DNA-origami samples. They then illustrate the method's performance by examining the oligomerization state (% of dimers vs % of tetramers) of the epidermal growth factor receptor (EGFR) under resting conditions and following EGF treatment. Furthermore, they investigate the hetero- and homodimerization states of two surface proteins involved in immune cell signaling (CD80 and PD-L1). Notably, the authors provide an open-source Python software package with a user-friendly and well-documented graphical interface, which will integrate seamlessly with their Picasso platform.

The prospect of synergizing the exceptional capabilities of DNA-PAINT and/or RESI to achieve single-protein resolution images with a robust quantitative framework to determine the stoichiometry and spatial dimensions of oligomers is indeed noteworthy. While many groups in the single-molecule localization microscopy (SMLM) community have used Nth nearest-neighbor distance distributions to infer clustering (Front. Bioinform. 1:724127, 2021), developed clustering algorithms (Nat. Comm., 11, 1493, 2020), or determined degrees of proteins co-localization (Sci. Adv. 2020, 6, eaay7193), few have addressed the geometrical organization of proteins forming higher-order oligomers. In fact, the authors themselves previously introduced quantitative single-molecule colocalization analysis (qSMCL) in Nat. Commun. 12, 919, 2021. That method incorporated key elements now extended in SPINNA, such as labeling efficiency (LE), in silico data simulation, and Nth nearest-neighbor distance distributions, to determine fractions and spatial coordinates of interacting species, especially in heterogeneous protein distributions.

Given this context, SPINNA could have been anticipated to perform well in biologically relevant scenarios where oligomers form higher-order clusters or align along linear structures. This is particularly important as many oligomeric proteins in immunology, cancer biology, and neurology (to name a few) do not distribute randomly but instead organize in clusters or along linear structures. Without addressing these scenarios, the excitement within the community for SPINNA's current implementation may be limited. The following points highlight key areas for improvement and additional clarification:

(1) In the "Concept and Algorithm" section, the authors emphasize that labelling efficiency (LE) significantly influences SPINNA's accuracy in determining the number of oligomeric states and their spatial distribution. However, while the authors recently introduced a method for determining labelling efficiency (Nat Methods, 21, 1702-1707, 2024), the practical implementation of this experiment may present challenges depending on the biomolecule or target under investigation. Furthermore, even if the method is implemented, the error in determining labelling efficiency is reported to be in the range of 10–20%. It would be highly beneficial for the community if the authors could provide quantitative insights into how a 10–20% error in LE affects SPINNA's accuracy in determining oligomeric states and spatial distributions. On the other hand, given that several tested antibodies in the authors' Nat Methods 2024 study exhibited LE values in the range of 40–50%, it would be helpful to clarify whether these values could serve as reasonable initial estimates for analysis when LE measurements are not available. Such guidance would be particularly beneficial for researchers encountering difficulties in determining LE experimentally.

(2) The authors demonstrate in Supplementary Figure 1 that using the NND distributions for CSR calculated with a cell-specific mask, rather than relying on an average protein density value, enhances the accuracy of determining the percentage of oligomeric states and protein distributions. However, generating a mask requires users to define parameters such as Gaussian blur and mask threshold. It would be highly beneficial if the authors could provide quantitative insights into how the accuracy of the results depends on these parameters. In this lines, can the authors elaborate on what's the impact on SPINNA's accuracy when increasing the density origami samples where using a mask might not be applicable.

(3) The authors should also provide a clearer explanation of how users can determine the appropriate granularity for calculations and decide whether this parameter needs to be increased or whether alternative numbers of molecular structures should be considered to adequately describe their experimental data. On these lines, it would be highly beneficial if the authors could include information on how the computational time scales with the size of the search space and the granularity of the analysis.

(4) Indeed, this reviewer has tested the SPINNA module with experimental data and observed that the goodness of the fit is highly sensitive to the model distances between the molecular targets of the simulated structures and the different numbers of simulated structures. To aid in defining the model structures, the authors should clarify whether the information derived from the first maxima of the 1st nearest-neighbor distance (NND) distribution can be used as an a priori estimate for determining the minimal distances between proteins in dimers, trimers, tetramers, and other oligomeric states. Providing such guidance would help streamline the modeling process when there is little structural information available on the target under investigation.

(5) Furthermore, considering that different combinations of the search space could provide good levels of confidence, it would be valuable if the authors can include the Kolmogorov-Smirnov 2-sample test statistic (KS2) values in the supporting information for both simulations and experimental data. Presenting these values in a confusion matrix format could help users understand whether multiple configurations of the search space yield similar confidence levels for the output model. Additionally, the authors should discuss strategies for distinguishing between such cases, as this would be critical for interpreting the results.

(6) On a related topic, the authors should explicitly explain how specific distances were chosen/calculated for defining model structures in the analysis of their experimental data (e.g., 21 nm for EGFR dimers and tetramers; 10 nm, 15 nm, and 11 nm for CD80, PD-L1, and CD80–PD-L1 dimers, respectively). Adding references to justify these distances would strengthen the models.

(7) In the “Search Space” subsection of the “Concept and Algorithm” section, the authors briefly mention that the search space can be expanded to account for localization/label position uncertainty and/or intermolecular distances. This point warrants further elaboration in the manuscript, ideally supported by analysis using their experimental datasets (e.g., CD80, PD-L1, or EGFR). For example, it would be valuable to determine whether their approach can recover the 21 nm dimer distance of EGFR.

More generally, the authors should clarify the capabilities and limitations of SPINNA in determining spatial and structural information when little prior knowledge is available about the dimensions of protein oligomers. Providing such insights would underscore the utility of SPINNA, particularly when paired with the angstrom-level resolution achievable using RES1, for advancing structural biology investigations. An analytical framework capable of addressing these challenges is notably scarce in the literature and would be a significant contribution to the field.

(8) In the discussion section, the authors mention that SPINNA could be extended to different models with more complex structures (flexible chain of oligomers with variable lengths, clusters of molecules), etc. This capability is critical as many proteins form higher-order clusters in biological contexts. From this reviewer's testing, SPINNA seemed to perform poorly with datasets that contained oligomers inside higher-order clustered regions, yet it seems feasible to integrate clustering metrics (e.g., Ripley's K function) into the analysis. The authors should present examples of the performance of SPINNA in such cases and implement a suitable modification to account for these cases, as clustering is extremely common in relevant biological targets under investigation.

(9) In the discussion section, the authors also mention that for hypothesis-free scenarios, SPINNA should be complemented with other approaches, such as those based on spatial clustering and point-pattern data mining. It would be helpful if the authors could expand on this statement, providing more details to clarify how these complementary methods could be integrated with SPINNA.

(10) In the Methods section, the authors should explicitly state how uncertainties in protein positions were calculated for the experimental datasets via their SMLM clustering algorithm. Furthermore, they should clarify whether SMLM clustering on DNA-PAINT can recover single-protein positions within higher-order clusters below the DNA-PAINT resolution limit, potentially leveraging qPAINT where appropriate.

(Remarks on code availability)

The authors have developed an open-source Python software package featuring a user-friendly and well-documented graphical interface. Upon testing the GUI, this reviewer successfully reproduced the analyses presented in the manuscript. Additionally, the README file was exceptionally clear and straightforward, making the setup process seamless.

Reviewer #2

(Remarks to the Author)

In the submitted manuscript entitled “Spatial and stoichiometric in situ analysis of biomolecular oligomerization at single protein resolution” by Masullo LA et al., the authors developed an analysis method, SPINNA, to compare nearest-neighbor distances from experimental single-protein positions data with the distances obtained from simulated data based on a model of protein oligomerization states, and confirmed the molecular behavior of the three representatives of biologically important molecules, structure in DNA origami, EGFP bound-induced homodimerization of EGFR and heterodimerization of CD80 and PD-L1. Furthermore, it is important to provide an open-source Python implementation and a GUI for usage in the scientific community.

These results imply strategical advances and new insights into understanding the facts of biophysics and biology and contains beneficial interests both fields. This analysis method is quite excellent, but it contains some critical issues to be solved particularly to fit the models to biology, so the reviewer demands some significant improvements in the experiments and the manuscripts. Most of the issues arise from whether the densities of the molecules on the cell surface being observed is within the physiological ranges because they were artificially expressed for analyses. Also, by comparing cells with high and low EGFR expressions, could the authors show that data in Figure 3 changes consistently?

1. In Figure 3, authors showed that EGFR, expressed as a monomer on the cell surface, forms a homodimer and even a tetramer in the presence of the ligand EGF. However, even in the absence of ligands, some EGFRs are known to be expressed as a homodimer, and the binding to EGFP is known to introduce intracellular structural changes in the EGFR homodimers for inducing their signaling. It has also been reported that the proportion of EGFR oligomers varies depending on the density of EGFR on the cell surface (doi: 10.1038/nature08827). Are the proportions of monomer, dimer and tetramer of oligomers in Figure 3i reasonable compared to those proportions reported so far?

2. In Figure 4, the authors showed the proportions of monomers, homodimers, and heterodimers in each molecule when both CD80 and PD-L1 are simultaneously expressed in a cell. The authors suggested that PD-L1 expression was lower than CD80, which is why many CD80s form dimers and 35% of PD-L1 forms monomers. Since it is biologically known that CD80 is naturally expressed as a homodimer and PD-L1 is expressed as a monomer, it is not desirable that excessive expression of PD-L1 affects the biological dynamics of PD-L1, which is originally expressed as a monomer. To reduce the PD-L1 expression is one of the methods to overcome that issue, and the reviewer believes another way. CD86 is known to be expressed as a monomer, and the cis binding of CD86 to PD-L1 has not been reported. Therefore, if the authors show any differences in such proportions between CD80 and CD86 by comparing the CD80-PD-L1 and CD86-PD-L1 pairs, they could solve the issue to some extent. In addition, the reviewer would like to know the proportions of each molecule if CD80 or PD-L1 is expressed alone.

(Remarks on code availability)

Version 1:

Reviewer comments:

Reviewer #1

(Remarks to the Author)

The authors have thoroughly addressed all the reviewers' concerns, including mine. They have provided additional supplementary analysis to support previously unclear statements, enhanced the explanations for certain figures, and incorporated new text to directly respond to my initial concerns. I fully support recommending this paper for publication in Nature Communications.

(Remarks on code availability)

The provided code is well-documented and includes a README file with clear instructions for installation and execution. I was able to successfully install and run the application without issues.

Reviewer #2

(Remarks to the Author)

The author answered the reviewer's questions very well on the technical aspects.

(Remarks on code availability)

REVIEWER COMMENTS

Reviewer #1 (Remarks to the Author):

In this manuscript, Masullo et al. report on SPINNA (Single-Protein Investigation via Nearest-Neighbor Analysis), a model-based method capable of determining the stoichiometry and the spatial arrangement of molecular targets within cells from single-protein resolution images, as well as calculating the percentage of molecular targets that adopt different geometrical configurations (i.e., linear vs triangular, dimers vs tetramer, etc). The authors benchmark the method's capabilities using both simulated data sets and DNA-origami samples. They then illustrate the method's performance by examining the oligomerization state (% of dimers vs % of tetramers) of the epidermal growth factor receptor (EGFR) under resting conditions and following EGF treatment. Furthermore, they investigate the hetero- and homodimerization states of two surface proteins involved in immune cell signaling (CD80 and PD-L1). Notably, the authors provide an open-source Python software package with a user-friendly and well-documented graphical interface, which will integrate seamlessly with their Picasso platform.

The prospect of synergizing the exceptional capabilities of DNA-PAINT and/or RESI to achieve single-protein resolution images with a robust quantitative framework to determine the stoichiometry and spatial dimensions of oligomers is indeed noteworthy. While many groups in the single-molecule localization microscopy (SMLM) community have used Nth nearest-neighbor distance distributions to infer clustering (Front. Bioinform. 1:724127, 2021), developed clustering algorithms (Nat. Comm., 11, 1493, 2020), or determined degrees of proteins co-localization (Sci. Adv. 2020, 6, eaay7193), few have addressed the geometrical organization of proteins forming higher-order oligomers. In fact, the authors themselves previously introduced quantitative single-molecule colocalization analysis (qSMCL) in Nat. Commun. 12, 919, 2021. That method incorporated key elements now extended in SPINNA, such as labeling efficiency (LE), in silico data simulation, and Nth nearest-neighbor distance distributions, to determine fractions and spatial coordinates of interacting species, especially in heterogeneous protein distributions.

We thank the reviewer for their appreciation of our work.

Given this context, SPINNA could have been anticipated to perform well in biologically relevant scenarios where oligomers form higher-order clusters or align along linear structures. This is particularly important as many oligomeric proteins in immunology, cancer biology, and neurology (to name a few) do not distribute randomly but instead organize in clusters or along linear structures. Without addressing these scenarios, the excitement within the community for SPINNA's current implementation may be limited. The following points highlight key areas for improvement and additional clarification:

(1) In the "Concept and Algorithm" section, the authors emphasize that labelling efficiency (LE) significantly influences SPINNA's accuracy in determining the number of oligomeric states and their spatial distribution. However, while the authors recently introduced a method for determining labelling efficiency (Nat Methods, 21, 1702-1707, 2024), the practical implementation of this experiment may present challenges depending on the biomolecule or target under investigation. Furthermore, even if the method is implemented, the error in determining labelling efficiency is reported to be in the range of 10–20%. It would be highly beneficial for the community if the authors could provide quantitative insights into how a 10–20% error in LE affects SPINNA's accuracy in determining oligomeric states and spatial distributions.

We have now added a quantitative evaluation of how the uncertainty in the LE determination impacts the overall uncertainty of the SPINNA analysis. Supplementary Tables 1 and 2 now display the results of the SPINNA analysis for three values of LE: "mean - std", "mean", and "mean + std" according to the uncertainties with which we measured the LE of the anti-GFP nanobody (for EGFR-mEGFP in Figure 3 and PD-L1-mEGFP in Figure 4) and anti-mCherry nanobody (for CD80-mCherry in Figure 4). We have also added a sentence discussing this in the main text of the manuscript.

On the other hand, given that several tested antibodies in the authors' Nat Methods 2024 study exhibited LE values in the range of 40–50%, it would be helpful to clarify whether these values could serve as reasonable initial estimates for analysis when LE measurements are not available. Such guidance would be particularly beneficial for researchers encountering difficulties in determining LE experimentally.

In general, all the LE values reported in our previous work (Hellmeier, Strauss, et al, Nature Methods 2024, doi:10.1038/s41592-024-02242-5) are good measures for LEs and could be used as a reference without further experiments. Binders with similar structures should perform similarly, but this cannot be guaranteed. We believe that in general for binders that have not been characterized before at the single-molecule level, it is not safe to assume a specific value (or a range) for LE. It is very important to perform a direct measurement and characterization of LE.

(2) The authors demonstrate in Supplementary Figure 1 that using the NND distributions for CSR calculated with a cell-specific mask, rather than relying on an average protein density value, enhances the accuracy of determining the percentage of oligomeric states and protein distributions. However, generating a mask requires users to define parameters such as Gaussian blur and mask threshold. It would be highly beneficial if the authors could provide quantitative insights into how the accuracy of the results depends on these parameters.

We have now added a panel to Figure S1 (panel **Fig. S1e**) that shows how the masking parameters affect the results of the analysis. As displayed in the accuracy matrices, SPINNA performs robustly over a large range of masking parameters.

In this lines, can the authors elaborate on what's the impact on SPINNA's accuracy when increasing the density origami samples where using a mask might not be applicable.

The reviewer is right in pointing out that for a system such as a DNA origami sample a mask is not applicable since the surface density should be approximately homogeneous and uniform across the coverslip used. Regarding how an increased (uniform) density would impact the results in a sample such as the DNA origami one, we expect SPINNA to perform robustly as it consistently takes density into account in the modeling. We have performed simulations at increasing density and obtained robust results for a wide range of densities for cases that are realistically similar to the DNA origami sample. The results are included now in an expanded version of **Supplementary Figure 4**.

(3) The authors should also provide a clearer explanation of how users can determine the appropriate granularity for calculations and decide whether this parameter needs to be increased or whether alternative numbers of molecular structures should be considered to adequately describe their experimental data. On these lines, it would be highly beneficial if the authors could include information on how the computational time scales with the size of the search space and the granularity of the analysis.

We have now added a paragraph giving a clearer explanation of how users can determine the appropriate granularity. Briefly, granularity determines the “percentage steps” (e.g. steps of 1%pt) for the proportions of oligomers that will be used to create the search space and thus it will limit the resolution in terms of oligomers proportions that can be evaluated, i.e. the computational resolution. When the search space is created and saved, the user can examine what the different resulting percentage steps are for every tested oligomeric structure and thus choose the appropriate granularity.

In principle, higher granularity does not have any disadvantage and always a sufficiently high granularity should be used. The only disadvantage is the computational cost that scales linearly with the number of points in the search space (numerical simulations performed). The number of points in the search space and the estimated total calculation time are displayed before the user decides to start the analysis.

All in all, we would recommend using a granularity that results in a “percentage step” that is substantially smaller than the smallest uncertainty coming from experimental sources (most importantly cell-to-cell variability). In concrete terms, we believe that a granularity resulting in percentage steps of 1%pt should be a robust choice for most common scenarios, but this should be evaluated on a per-case basis.

(4) Indeed, this reviewer has tested the SPINNA module with experimental data and observed that the goodness of the fit is highly sensitive to the model distances between the molecular targets of the simulated structures and the different numbers of simulated structures. To aid in defining the model structures, the authors should clarify whether the information derived from the first maxima of the 1st nearest-neighbor distance (NND) distribution can be used as an a priori estimate for determining the minimal distances between proteins in dimers, trimers, tetramers, and other oligomeric states. Providing

such guidance would help streamline the modeling process when there is little structural information available on the target under investigation.

We have now clarified in the manuscript that the position of the NND peaks can indeed be used as an *a priori* estimate for determining the minimal distances between protein oligomers when there is limited structural information available on the target under investigation. We have now expanded the SPINNA user interface to include different models (e.g. different distances between the molecular targets) as parameters for the fitting. The user guide now includes a detailed explanation of this new feature.

(5) Furthermore, considering that different combinations of the search space could provide good levels of confidence, it would be valuable if the authors can include the Kolmogorov-Smirnov 2-sample test statistic (KS2) values in the supporting information for both simulations and experimental data. Presenting these values in a confusion matrix format could help users understand whether multiple configurations of the search space yield similar confidence levels for the output model. Additionally, the authors should discuss strategies for distinguishing between such cases, as this would be critical for interpreting the results.

We have now included all the KS2 values in the supporting information for both simulations and our experimental data. We note that we also provide an uncertainty for the KS2 (Δ KS2) that can be used to compare different results (within the same SPINNA analysis) and check whether two results could or could not be interpreted as equally likely. We note also that SPINNA outputs these values automatically so users will have access to all this information when using the software for their own data.

(6) On a related topic, the authors should explicitly explain how specific distances were chosen/calculated for defining model structures in the analysis of their experimental data (e.g., 21 nm for EGFR dimers and tetramers; 10 nm, 15 nm, and 11 nm for CD80, PD-L1, and CD80–PD-L1 dimers, respectively). Adding references to justify these distances would strengthen the models.

The procedure by which we obtained those distances was the following: we first determined a range of realistic distances for our model, using previous studies in literature as a reference. For EGFR we used Peckys et al, Scientific Reports 2013 (doi:10.1038/srep02626), and for CD80 and PD-L1 we used Maurer et al, Nature Communication 2022 (doi:10.1038/s41467-022-29286-5), Stamper et al, Nature 2001 (doi:10.1038/35069118) and Chen et al, Protein & Cell 2010 (doi:10.1007/s13238-010-0022-1). The information in those papers was used as prior knowledge to estimate the range of plausible distances of the dimers of molecules considering the labels used, fluorescent protein tag, and nanobody, which considerably enlarge the measured distance. Still, the precise distance value was left as a free parameter and fitted in the optimization process. We note that the distances found in the literature are not expected to be the same as found by the optimization of our SPINNA model since in our case we must take into account the labels of the proteins. Our results are indeed consistent with the ones found in the literature if the label size is accounted for. We have now added the references to justify these ranges of distances. We also have clarified in the manuscript that distances were swept as another parameter to find the optimal value in each case.

(7) In the “Search Space” subsection of the “Concept and Algorithm” section, the authors briefly mention that the search space can be expanded to account for localization/label position uncertainty and/or intermolecular distances. This point warrants further elaboration in the manuscript, ideally supported by analysis using their experimental datasets (e.g., CD80, PD-L1, or EGFR). For example, it would be valuable to determine whether their approach can recover the 21 nm dimer distance of EGFR.

We apologize for not having been clear on this point. The 21 nm dimer distance of EGFR was actually determined in that way: i.e., sweeping over a set of different dimer distances and choosing the optimal value. We have now clarified this point and also added the geometrical parameters as (optional) user inputs in the user interface of SPINNA.

More generally, the authors should clarify the capabilities and limitations of SPINNA in determining spatial and structural information when little prior knowledge is available about the dimensions of protein oligomers. Providing such insights would underscore the utility of SPINNA, particularly when paired with the angstrom-level resolution achievable using RESI, for advancing structural biology investigations. An

analytical framework capable of addressing these challenges is notably scarce in the literature and would be a significant contribution to the field.

From our perspective, SPINNA as a framework for analysis should be used to test different models (hypotheses) that describe the observed data (positions of the biomolecules in the cell at single-protein resolution). When little prior knowledge is available, we would recommend performing some prior spatial analysis of the data (which we would conceptually keep separate from SPINNA) to build one or several plausible models that could explain the observations. We discuss specific methods of prior analysis in our answer to point (9). We would like to stress, however, that from a practical point of view, it is very often the case that there is indeed some prior information from previous biochemical or structural data that helps build a hypothesis on possible interactions between receptors, other kinds of proteins, or more generally, biomolecules of interest. Furthermore, even when there is no experimental information available, recent deep-learning methods such as AlphaFold ([doi:10.1038/s41586-021-03819-2](https://doi.org/10.1038/s41586-021-03819-2)) or Rosetta ([doi:10.1126/science.abj8754](https://doi.org/10.1126/science.abj8754)) could be used to predict oligomerization dynamics and conformations.

(8) In the discussion section, the authors mention that SPINNA could be extended to different models with more complex structures (flexible chain of oligomers with variable lengths, clusters of molecules), etc. This capability is critical as many proteins form higher-order clusters in biological contexts. From this reviewer's testing, SPINNA seemed to perform poorly with datasets that contained oligomers inside higher-order clustered regions, yet it seems feasible to integrate clustering metrics (e.g., Ripley's K function) into the analysis. The authors should present examples of the performance of SPINNA in such cases and implement a suitable modification to account for these cases, as clustering is extremely common in relevant biological targets under investigation.

Highly clustered areas can be treated in two ways:

(1) by accounting for them as a high-density area of oligomers using the density mask.

(2) by modeling more complex structures such as oligomer flexible chains or higher-order oligomers (molecular clusters)

Option (1) would be the preferred way if the molecules are hypothesized to form low-order oligomers (e.g. up to tetramers) but then in turn are clustered by some other reasons independent of the modeled molecules (e.g. endocytosis). Option (2) would be the preferred way if the molecules are hypothesized to directly form high-order oligomers or clusters. Option (1) is readily available in SPINNA and can work robustly. Option (2) has not been yet implemented, although it should be straightforward to include such complex models for users with intermediate programming skills. As a proof of concept, we confidentially include data and analysis from another manuscript currently in revision where we used an extension of SPINNA and modeled the oligomerization of CD20 upon treatment with the therapeutic antibody Rituximab using a flexible chain of molecules. The confidential data is contained in the file "CD20_chains_for_review.pdf".

(9) In the discussion section, the authors also mention that for hypothesis-free scenarios, SPINNA should be complemented with other approaches, such as those based on spatial clustering and point-pattern data mining. It would be helpful if the authors could expand on this statement, providing more details to clarify how these complementary methods could be integrated with SPINNA.

We have now added a paragraph discussing this point in deeper detail. Briefly, if no prior information is known about the oligomerization of the protein(s) of interest (hypothesis-free scenario) SPINNA will not provide a model to analyze. Rather, the user should build a plausible model from the data. Hence, spatial clustering and point-pattern data mining should be performed on the protein position data before SPINNA analysis. Properties of the NND distributions (e.g. peaks) can be used as proxies for interactions. Statistical functions like Ripley's K function could provide information on the sizes of the clusters. Segmentation algorithms such as DBSCAN can then be used to define the molecular clusters, and give a hint on the degree of oligomerization. Also, geometrical descriptors (shape, concavity, linearity, circularity, etc) of the clusters coming from the DBSCAN analysis could be used to point towards a suitable model to use in the downstream SPINNA analysis. A paragraph discussing these points has been added to the Discussion section of the manuscript.

(10) In the Methods section, the authors should explicitly state how uncertainties in protein positions were calculated for the experimental datasets via their SMLM clustering algorithm. Furthermore, they

should clarify whether SMLM clustering on DNA-PAINT can recover single-protein positions within higher-order clusters below the DNA-PAINT resolution limit, potentially leveraging qPAINT where appropriate.

The uncertainty in protein positions is estimated from the label size and accounts mainly for the variability in the orientation and positioning of the label (for example in our case, the fluorescent protein as a molecular tag plus the nanobody). The uncertainty deriving from the localization uncertainty after clustering localizations into single protein positions is, in our case, much smaller than the uncertainty coming from the label size (less than 1 nm vs 5-10 nm). DNA-PAINT performed at 2 to 3 nm localization precision, which results in approx. 5-7 nm resolution (as for the data presented in Figures 2, 3, and 4) can effectively resolve single protein positions for the proteins used in this study, and we believe generally 5-7 nm resolution could be safely regarded as single-protein resolution for most applications. If, for different experimental reasons, DNA-PAINT at 2-3 nm localization precision is not achievable, or the distances of interest are below 5 nm, RESI can be used for direct single-protein resolution measurements. Alternatively, DNA-PAINT in combination with qPAINT could be leveraged as demonstrated by Simoncelli et al (Cell Reports, 2020). We have now expanded the discussion in this direction in our manuscript.

Reviewer #1 (Remarks on code availability):

The authors have developed an open-source Python software package featuring a user-friendly and well-documented graphical interface. Upon testing the GUI, this reviewer successfully reproduced the analyses presented in the manuscript. Additionally, the README file was exceptionally clear and straightforward, making the setup process seamless.

We thank the reviewer for their appreciation and their feedback. We put a lot of effort into making all of our algorithms and codes open-source and user-friendly to maximize the impact on the scientific community.

Reviewer #2 (Remarks to the Author):

In the submitted manuscript entitled “Spatial and stoichiometric in situ analysis of biomolecular oligomerization at single protein resolution” by Masullo LA et al., the authors developed an analysis method, SPINNA, to compare nearest-neighbor distances from experimental single-protein positions data with the distances obtained from simulated data based on a model of protein oligomerization states, and confirmed the molecular behavior of the three representatives of biologically important molecules, structure in DNA origami, EGFP bound-induced homodimerization of EGFR and heterodimerization of CD80 and PD-L1. Furthermore, it is important to provide an open-source Python implementation and a GUI for usage in the scientific community.

We thank the reviewer for their appreciation of our work.

These results imply strategical advances and new insights into understanding the facts of biophysics and biology and contains beneficial interests both fields. This analysis method is quite excellent, but it contains some critical issues to be solved particularly to fit the models to biology, so the reviewer demands some significant improvements in the experiments and the manuscripts. Most of the issues arise from whether the densities of the molecules on the cell surface being observed is within the physiological ranges because they were artificially expressed for analyses. Also, by comparing cells with high and low EGFR expressions, could the authors show that data in Figure 3 changes consistently?

We have observed no correlation between the EGFR oligomerization and the EGFR expression, up to a surface density of 200 molecules per \$\mu\text{m}^2\$. These data will be included in a specific EGFR study that is currently under preparation. We confidentially include the preliminary data for the reviewer to scrutinize in the file “EGFR_density_for_review.pdf”.

1. In Figure 3, authors showed that EGFR, expressed as a monomer on the cell surface, forms a homodimer and even a tetramer in the presence of the ligand EGF. However, even in the absence of ligands, some EGFRs are known to be expressed as a homodimer, and the binding to EGFP is known to introduce intracellular structural changes in the EGFR homodimers for inducing their signaling. It has also been reported that the proportion of EGFR oligomers varies depending on the density of EGFR on the cell

surface (doi: 10.1038/nature08827). Are the proportions of monomer, dimer and tetramer of oligomers in Figure 3i reasonable compared to those proportions reported so far?

The proportions of monomers, dimers, and tetramers that we observed in our experiments are consistent with the values reported by Huang et al (Fig. 3a and Fig. 4b of eLife 2016, doi:10.7554/eLife.14107) and the values reported by Needham et al (Fig. 3e of Nature Communications 2016, doi:10.1038/ncomms13307). Differences in absolute values may arise from the usage of different cell lines, different treatment doses and times, and different measurement and quantification methods.

On the other hand, Chung et al (10.1038/nature08827) infer the dimerization of EGFR from diffusion measurements performed using single-molecule tracking. They measure the total dimerization of EGFR, they do not rule out heterodimerizations together with homodimerizations. Therefore their “dimerization probabilities” (which account for both homo and hetero interactions) are expected to be higher than the homodimerizations that we measured.

2. In Figure 4, the authors showed the proportions of monomers, homodimers, and heterodimers in each molecule when both CD80 and PD-L1 are simultaneously expressed in a cell. The authors suggested that PD-L1 expression was lower than CD80, which is why many CD80s form dimers and 35% of PD-L1 forms monomers. Since it is biologically known that CD80 is naturally expressed as a homodimer and PD-L1 is expressed as a monomer, it is not desirable that excessive expression of PD-L1 affects the biological dynamics of PD-L1, which is originally expressed as a monomer. To reduce the PD-L1 expression is one of the methods to overcome that issue, and the reviewer believes another way. CD86 is known to be expressed as a monomer, and the cis binding of CD86 to PD-L1 has not been reported. Therefore, if the authors show any differences in such proportions between CD80 and CD86 by comparing the CD80-PD-L1 and CD86-PD-L1 pairs, they could solve the issue to some extent. In addition, the reviewer would like to know the proportions of each molecule if CD80 or PD-L1 is expressed alone.

We agree that the higher expression of PD-L1, $(409 \pm 46) \mu\text{m}^2$, as compared to CD80, $(142 \pm 39) \mu\text{m}^2$, could generate oligomerization dynamics that are not found in cells expressing these molecules at physiological expression levels. However, the focus of our experiment was to show that we are able to quantify the oligomerizations using SPINNA in this system, and as such we believe our experiment to be a solid demonstration of our algorithm implementation. In order to draw biologically relevant conclusions

we agree that the expression level of PD-L1 should be reduced substantially. This could be achieved by engineering a suitable cell line with reduced PD-L1 expression. Similarly, additional cell lines could be engineered to express only CD80 or PD-L1. We foresee that the oligomerization dynamics will in fact change in such cell lines as we observe a cis heterodimerization between CD80 and PD-L1 that would disappear in the scenarios in which we express CD80 or PD-L1 only. We respectfully believe that such experiments are out of the scope of this manuscript which is to describe our quantification method, SPINNA, and proof-of-principle applications. However, we have now added a paragraph discussing this as a potential outlook.